# An Economic Evaluation of ‘Sheds for Life’: A Community-Based Men’s Health Initiative for Men’s Sheds in Ireland

**DOI:** 10.3390/ijerph19042204

**Published:** 2022-02-15

**Authors:** Aisling McGrath, Niamh Murphy, Tom Egan, Gillian Ormond, Noel Richardson

**Affiliations:** 1School of Health Sciences, Waterford Institute of Technology, X91 K0EK Waterford, Ireland; nmurphy@wit.ie; 2School of Business, Waterford Institute of Technology, X91 K0EK Waterford, Ireland; tegan@wit.ie (T.E.); gormond@wit.ie (G.O.); 3National Centre for Men’s Health, Institute of Technology Carlow, R93 V960 Carlow, Ireland; noel.richardson@itcarlow.ie

**Keywords:** men’s health, economic evaluation, cost-effectiveness, community, men’s sheds

## Abstract

Men’s Sheds (‘Sheds’) attract a diverse cohort of men and, as such, have been identified as spaces with the potential to engage marginalized subpopulations with more structured health promotion. ‘Sheds for Life’ is a 10-week men’s health initiative for Sheds in Ireland and the first structured health promotion initiative formally evaluated in Sheds. Cost is an important implementation outcome in the evaluation of Sheds for Life when operating in an environment where budgets are limited. Therefore, an economic evaluation is critical to highlight cost-effectiveness for decision makers who determine sustainability. This is the first study to evaluate the cost-effectiveness of health endeavors in Sheds. All costs from pre-implementation to maintenance phases were gathered, and questionnaires incorporating the SF-6D were administered to participants (*n* = 421) at baseline, 3, 6, and 12 months. Then, utility scores were generated to determine quality-adjusted life years (QALYS). Results demonstrate that the intervention group experienced an average 3.3% gain in QALYS from baseline to 3 months and a further 2% gain from 3 months to 6 months at an estimated cost per QALY of €15,724. These findings highlight that Sheds for Life is a cost-effective initiative that effectively engages and enhances the well-being of Shed members.

## 1. Introduction

Traditionally, men have been regarded as being more difficult to engage with conventional health services compared to women, and an understanding of how gender shapes men’s health practice is a critical first step in developing effective health promotion strategies that might appeal to men [1]. Indeed, the importance and success of gendered approaches in the design and delivery of health interventions for men has been highlighted in a host of community-based men’s health programs [2,3,4,5,6]. These approaches also demonstrate a need for a more targeted approach to recruit more marginalized groups of men [5]. Research spanning Australia, Ireland, and the UK has cemented the reputation of Men’s Sheds (‘Sheds’) as settings that are inherently health promoting for men, with Sheds increasingly being seen by health and social policy makers as an exemplar for the promotion of men’s health and well-being [7,8,9,10]. The Men’s Shed movement was first founded in Australia in the 1980s and has since expanded to other countries, first arriving in Ireland in 2011 and growing exponentially with over 450 Sheds now on the island and up to 10,000 members. Sheds are community-based independent and self-autonomous where men come together of their own volition to socialize in the company of other men. The exponential and organic growth of Sheds has been highlighted as a testament to a need for men to identify with an environment that offers a sense of safety and purpose [7]. Sheds engage in a range of activities, such as woodwork, music, and community outreach that foster opportunities to participate in meaningful activities that encourage skill sharing, informal learning, camaraderie, and belonging facilitated within a socially acceptable and masculine environment [7,9,10]. Sheds operate on minimal funding and are self-sustained. The Irish Men’s Sheds Association (IMSA) supports the development of the network of Sheds in Ireland. The inherent health promotion qualities of Sheds such as the sense of purpose, meaning, and social support offered within them make the Sheds highly conducive to health promotion endeavors [11,12,13]. Moreover, because they are community-based and non-clinical environments, research has found that Sheds typically attract more vulnerable subpopulations of ‘hard-to-reach’ (HTR) groups of men—older, more marginalized male subpopulations, who typically might not otherwise engage with health services or programs [10,11,14]. Thus, policy makers and researchers have called for structured health promotion endeavors in Sheds, querying what this might look like and how it might be effectively delivered without compromising the integrity of Sheds [7,10,12,15]. However, to date, there remains limited high-quality or empirical research evidencing the links between Sheds and health and well-being, which has been a noted limitation in assessing the Shed–health relationship [7,10,16]. To our knowledge, there has been no other structured health promotion initiatives evaluated in Sheds nor has there been any economic evaluation of health promotion in Sheds.

### 1.1. Nature of Sheds for Life

The concept for Sheds for Life (SFL) was first developed in 2016 in response to a commitment from the Sheds representative body in Ireland (Irish Men’s Sheds Association; IMSA) to prioritize health initiatives for its membership. Since then, an alliance of stakeholders including the IMSA, academics, funders, policy makers, provider organizations, and importantly Shed members (‘Shedders’) themselves, have guided the evolution of SFL into a ten-week initiative that delivers targeted and tailored health promotion directly in the Sheds setting. A detailed description of the SFL intervention and evaluation approach is available in a protocol paper that outlines its design, implementation, and evaluation methods [12]. In short, SFL begins with a health check in the Sheds and then focuses on priority areas of healthy eating, physical activity, and mental health with additional optional components that allow Sheds to tailor the initiative to respond more accordingly to their needs, such as health awareness sessions on diabetes, cancer, dementia, and oral health, CPR, digital literacy, and suicide prevention training. Sheds for Life builds upon the informal, safe, and familiar environment of Sheds and employs gender-specific approaches to further enhance the adoption and reach of the initiative in Sheds [1]. While the informality of Sheds is an advantage in engaging men and needs to be respected in order to uphold the integrity of the Sheds [10], it presents challenges in terms of structured program delivery and evaluation. The Sheds by nature are highly variable, autonomous, non-structured spaces where attendance can be sporadic and where members are not compelled to undertake any activity. Therefore, the challenge is to develop a pragmatic delivery design that can operate within the organic, non-structured space of Sheds where contextual factors vary within and beyond Sheds in terms of the wider systems. For this reason, the broader evaluation of SFL utilizes a hybrid effectiveness-implementation design guided by implementation frameworks [17,18,19]. Central to this approach is the use of community-based participatory research methods where Shedders are key decision makers in the design and delivery of SFL in partnership with other key stakeholders. Indeed, a critical success factor for SFL is this partnership approach, where partner organizations understand the ethos of Sheds and recognize the value in engaging men with health. Moreover, SFL adopts a sustainable delivery model in that it is delivered under real-world conditions, where service provider organizations undertake SFL delivery as part of their routine work plans—as opposed to short-term (and often unsustainable) grant funding. That said, finite resources both in terms of a limited implementation workforce and competing priorities among provider organizations demand that a prudent approach is taken to matching Sheds’ needs with program offerings. This also highlights the importance of economic evaluation to determine the cost-effectiveness of SFL and inform the allocation of said finite resources. A detailed outline of this approach can be accessed in the SFL protocol [12].

### 1.2. Assessing Costs of Health Programs

The true cost impact of a particular intervention depends upon the implementation strategy used and the location of delivery [17]. Proctor et al. [17] outline that the measurement of implementation costs designed to demonstrate cost-effectiveness are essential for studies in real-word settings and appeal to both policy makers and funders. This is particularly relevant in the case where costs of the intervention may be compared with other alternative treatments or implementation strategies. Implementation costs associated with an intervention are also likely to impact the rating of acceptability of the intervention, and therefore, costs are important to measure in the remit of an implementation study where acceptability is sought to be understood [17]. For policy makers and funders, it will be important to demonstrate the cost-effectiveness of SFL when potentially allocating finite public funds, particularly as there is a lack of evidence pertaining to the impact of men’s health promotion in Sheds with no available research to date on the economic evaluation of health promotion in Sheds [10,15,20]. Studies investigating the economic impact of male health programs have been limited to date; however, strategies that seek to improve men’s health have been found to have cost-saving benefits [21]. For example, an economic assessment outlining the costs of men’s health disparities demonstrated that the premature morbidity and mortality of men sharply increased government and private sector expenditures [22]. This research demonstrated that men’s premature mortality and morbidity has been estimated to cost the United States economy approximately USD 479 billion annually. Krueger et al. [23] also assessed the economic impact of modest health behavior change in middle-aged men who smoke tobacco, consume excess alcohol, are physically inactive, and have excess weight—modifiable risk factors that cost upwards of 730.4 billion dollars in the US annually [24]. The research found that modeling a 1% annual relative reduction each year through to 2036 would result in a cumulative cost avoidance between the years 2013 and 2036 of 50.7 billion CAD. The research also determined that health interventions that can encourage a modest annual reduction in risk factors can have an important public health and cost-saving impact. For example, an economic evaluation of Men on the Move (a community-based physical activity program for middle-aged men) demonstrated the program to be cost-effective in support of an at-risk cohort of men with an estimated QALYs ratio cost of €3723, which is significantly less that the existing benchmark of €20,000 to €45,000 [25]. Notwithstanding the utility of Sheds in engaging more marginalized subpopulations of men, programs for Shed members also have the potential to be cost saving. Therefore, it is important that the cost-effectiveness of SFL is assessed as a critical component in highlighting the case for health promotion in Sheds. Moreover, Brott et al. [22] argue that “the social justice” argument (that saving men’s lives is simply the right thing to do) is not always enough to incite action; rather that research should focus on demonstrating the return on investment gained from engaging men with health services and programs at the prevention stage, and making a ‘business case’ for men’s health promotion that appeals to decision makers. Moreover, in an environment where budgets are limited with many programs all vying for funds, economic evaluations are not only beneficial but also a necessary tool to the decision-making process.

Quality-Adjusted Life Years are universally applicable as they amalgamate the impacts of interventions on both quality and quantity of life in a single, common metric, thus facilitating comparisons between different health programs [26]. The Health Information Quality Authority (HIQA) highlights the usefulness of this approach to decision makers with limited resources [26]. Indeed, QALYs are considered a cornerstone of economic analysis and aid decision making in healthcare, particularly regarding the prioritization of limited resources [27]. The incremental cost effectiveness ratio enables the cost of a program to be compared to known benchmarks to assess its effectiveness. Making use of cost-effectiveness ratios in relation to public health policies are very helpful in assessing such trade-offs [28]. This research sought to determine whether the SFL initiative was an effective model in terms of health improvements and cost outcomes considering the perspective of the health services [29]. Therefore, the purpose of this study was to conduct an economic evaluation of the SFL Program to (i) investigate if the SFL intervention was a cost-effective approach capable of improving health outcomes of participants; (ii) demonstrate the cost-effectiveness of SFL with a view to enhancing its acceptability among key stakeholders and decision makers; and (iii) highlight the benefit of economic evaluation for others engaged in men’s health and community-based health promotion. Quality-Adjusted Life Years along with the Incremental Cost Effectiveness Ratio will be calculated to enable the cost-effectiveness of the SFL initiative to inform scalability.

## 2. Materials and Methods

### 2.1. Study Participants

Following assessment of the implementation environment, namely the capacity and resource constraints of provider organizations to deliver SFL along with the nuances, ethos, and autonomy of the inner (Sheds) setting, the SFL 10-week intervention was implemented on a phased basis across two cohorts, each consisting of two counties in Ireland. The first program was delivered in Counties Kildare and Waterford between March and May 2019. The population of those counties is ca. 222,504 and 116,176, respectively [30]. The second program was delivered in Counties Limerick and Louth from September to November 2019, each with a population of ca. 194,899 and 128,884, respectively [30]. Whilst delivery occurred in the first cohort (*n* = 12 clusters; *n* = 212 Shedders), a wait list control cohort served as a comparator (*n* = 3 clusters; *n* = 89 Shedders), and these were a subset of the second cohort (*n* = 9 clusters, *n* = 209 Shedders). Purposive sampling was used to recruit Shedders to the SFL program and was carried out through an expression of interest process targeting each Shed and a series of Shed visits conducted by the research team and members of the IMSA. This recruitment strategy was used in line with the gendered approach of SFL as a key enabler to engagement. In total, *n* = 31 Sheds out of a potential *n* = 44 (70%) across the selected counties opted into SFL. Data were collected at the recruitment phase to identify the number of Shedders who regularly attended the participating Sheds to establish the reach of SFL. It was estimated that *n* = 565 were active members of the participating Sheds at the time of recruitment, with the majority (*n* = 421; 75%) opting to participate in SFL and the supporting evaluation, suggesting that the recruitment strategy was effective in engaging the target group. Inclusion criteria comprised all adult males who were active Shed members, had a good proficiency in the English language, and could give informed consent.

### 2.2. Data Collection

Program outcomes were reported through questionnaires that were administered and completed by each of the participants on a one-to-one basis with a member of the research team to account for potential literacy issues. The questionnaires assessed a range of measures of various lifestyle variables along with well-being and self-rated health, for each participant, and information on the participants was gathered at baseline, 3 months (following completion of the 10-week intervention), 6 months, and 12 months (see McGrath et al. [12] for further information on instruments used).

Costs of implementation and maintenance of the program were gathered by the research team and SFL delivery agencies across the two cohorts for up to 12 months. Both direct and indirect costs incurred in the implementation of the program were recorded in the period up to 3 months. Further maintenance costs of the program were recorded from 3 months to 12 months, which included costs borne by provider organizations to deliver different elements of SFL (health check, mental health workshop, cancer awareness, etc.); costs incurred by the IMSA to coordinate its delivery (administration, salaries, travel, and subsistence), alongside other miscellaneous costs (e.g., awards event for participants who completed SFL). For the purpose of this economic evaluation, costs were restricted to those incurred in the intervention element of the program, and the research costs that were incurred in the planning of the program were not included, as these costs would not be incurred during any subsequent delivery of SFL [31].

### 2.3. Methodological Approach

The SF-36 health survey is one of the most widely used measures of health-related quality of life [32]. The short form 6D (SF-6D) is a reduced form of the general health measure SF-36 and is widely recommended as a generic preference based method to measure utility [26]. It measures 6 dimensions of health: physical functioning, role limitations, social functioning, pain, mental health, and vitality, with each dimension having between two and six levels allowing for a potential 18,000 varying health states to be defined. Responses to the questionnaire are coded, with the codes then summed to produce a total score. The scores enable health differences between individuals or groups to be displayed and changes to health as a result of an intervention to be detected. Participants in SFL completed the SF-6D questionnaire at baseline, 3, 6, and 12 months, which allowed a six-digit health state code to be created for each individual at each of the different time points. Then, these were converted into utility weights using the SF-6D algorithm. In the absence of Irish public preference data, preference weights used in the SF-6D are obtained from a sample of the general population in the UK using the recognized valuation technique of standard gamble. A repeated-measures ANOVA and paired-samples *t*-tests were also performed to determine significant differences in utility scores across time points in the Intervention Group (IG) and Control Group (CG) groups, respectively.

The utility values generated from the SF-6D questionnaires in SFL allowed QALYs to be calculated. A QALY rate of 1.0 represents full health and 0.0 represents death [28]. The incremental cost-effectiveness ratio (ICER) is a means to assess the cost of a program relative to its effectiveness. In the context of this study, the ICER was derived to show the additional cost for one additional QALY gained by the Intervention Group (IG) compared to the CG. This enabled the SFL project to be assessed on the basis of its net benefit to the participants. Thresholds used in Ireland for cost effectiveness purposes can vary. For pharmaceutical interventions, a threshold of €45,000 per QALY is used; however, for non-drug interventions, HIQA state that the threshold used has tended to be between €20,000 and €45,000 [26,33]. This is broadly similar to the UK threshold where the National Institute of Health and Care Excellence sets the threshold at £20,000 [34]. On this basis, if the ICER for the SFL was to fall below €20,000 per QALY gained, this would be viewed as a strong endorsement for the project.

## 3. Results

The cost utility analysis conducted on SFL involved assessing the incremental costs and benefits of the program. Table 1 sets out the costs associated with the program, the majority of which relate to implementation costs in the initial three months. Costs corresponding to later time periods relate to maintenance of the program. The pre-implementation planning costs that relate to staffing for those involved in SFL delivery and some travel costs are included in the baseline to 3-month costs.

For the SFL program, the vast majority of the costs are related to the initial three-month period, and while some of these costs were significant such as salary costs and the costs of full health checks, this must be considered against the success of this program in attracting 421 participants from what is considered a ‘hard-to-reach’ population. This leads to a cost-effectiveness ratio (CER) of €309.3 per QALY, which can be considered in comparison to a ‘‘do nothing’’ scenario. In the case of Sheds, a ‘do nothing’ scenario would mean choosing not to deliver structured health promotion in Sheds as no other alternative has yet been explored for Sheds.

The incremental benefits of the program involved generating utility levels for each participant as per Brazier et al. [32]. Initially, results for the six components of SF-6D were generated to see how the various elements were rated by the participants over this period; these are shown in Table 2 below.

Table 2 demonstrates that a sample size of over 200 participants was generated at all time periods of this study, and this was after extensive efforts given the challenges of completing the study during parts of the COVID-19 pandemic [13]. In interpreting this table, note that for the first five dimensions (Physical Functioning to Mental Health), the scale used (from 1 to 5) scored healthier people with lower values, while for the last dimension (Vitality), healthier people were represented by higher scores. With this in mind, it can be seen that the mean values for the IG showed improvement across all dimensions from baseline to 3M (the highest improvement is in the physical functioning dimension), while most dimensions improved further from 3M to 6M before largely leveling off from 6M to 12M. This contrasts with the CG for which the data show only very modest changes in all six dimensions from baseline to 3M, after which no further data was available. As per Brazier et al. [32], the six components were then amalgamated to generate QALYs for IG and the CG, as shown in Table 3 below.

In terms of utility changes, the IG achieved a 3.3% gain in the first three months compared to just a 1% gain for the CG and achieved a further 2% gain in the following three months. This leads to a QALY gain of almost 12 (7.595 + 4.222) for the IG in the first six months with a slight reduction at 12 months when the benefits of SFL may have reduced. The COVID-19 pandemic is a potential confounder here that may in part explain the diminishing utility, and the impact of COVID-19 on SFL participants has been discussed elsewhere [13]. As SFL was delivered on a phased basis, Cohort 1 was followed up to 12 months prior to COVID-19 restrictions. However, Cohort 2 was actively experiencing COVID-19 restrictions during the 6 and 12 month follow-ups. A comparison of both cohorts using independent samples *t*-tests determined that there were no significant differences between these groups utility scores at all follow-up time points (*p* > 0.05). This suggests a limited correlation between COVID-19 and the trajectory of participant utility scores.

A repeated-measures ANOVA with a Bonferroni adjustment was performed to ascertain the significance between changes in utility scores across time points (see Table 3). Results determined there was a significant improvement in utility scores from baseline to all other time points (3, 6, and 12 months; f = 9.96, *p* = 0.000).

While constraints with the study design meant that there were no data available for the CG beyond 3M (see data collection and limitations sections), a paired samples *t*-test determined that there was no significant difference (*p* > 0.05) in this group from baseline to 3 months compared to the IG who received the SFL intervention. It is also notable that Shedders rated their baseline utility scores as relatively high in both the intervention (0.795) and control groups (0.777), which leaves less room to achieve further utility gains. This analysis culminates in a cost per QALY ratio of €15,724, which is highly cost effective when compared to the generally accepted thresholds of at least €20,000 in Ireland and the UK [33,34].

### Sensitivity Analysis

While the above analysis suggests that the SFL program is cost effective, this finding relates to a single study. Therefore, it is useful to ascertain a level of certainty about these values for the implementation of similar programs. Capturing uncertainty for multiple variables involves assessing standard deviation and confidence intervals; however, this is complicated when dealing with a ratio (incremental cost effectiveness ratio—ICER). A common approach to capturing uncertainty for ratio variables is a probability sensitivity analysis using Monte Carlo simulation, as this can lead to a cost-effectiveness acceptability curve (CEAC) [35]. In this study, 1000 different values for all cost and QALY values are generated, and this leads to a CEAC curve that plots the probability of cost effectiveness against different threshold values, as shown in Figure 1 below.

The probability sensitivity analysis is based on modeling the average cost and average QALY using a normal distribution. Based on the curve highlighted in Figure 1, if one is willing to pay at least €15,000 per QALY (a figure close to the estimated cost per QALY in this program), there is an 89% chance that this program is more effective compared to other programs. These data capture the potential uncertainly surrounding a resource allocation decision, and SFL is shown to have a high probability of being successful if one is willing to pay over €10,000 per QALY, which is below costs-effectiveness thresholds in the UK and Ireland [33,36]. For distribution of each SF-6D dimension see Appendix A which depicts histograms of the six SF-6D dimensions at baseline.

## 4. Discussion

This paper sought to conduct an economic evaluation of SFL, which is the first structured men’s health promotion program in the Shed setting [12]. Given the lack of formal evaluation of health promotion in Sheds, not surprisingly, there has been no formal economic evaluation of such endeavors, with research further highlighting a distinct lack of economic evaluation for men’s health initiatives and public health interventions more broadly [10,15,37,38]. Therefore, the findings fill an important gap in the literature by assessing the cost effectiveness of a tailored and gender-specific health promotion initiative (SFL) targeted at an HTR cohort of men in the Shed setting. Findings also build upon the recommendations of a previous community-based physical activity program designed for middle-aged men, Men on the Move, which highlights the efficacy of gender-specific, community-based men’s health initiatives that can effectively engage men and are also cost saving [25]. Moreover, advocates of implementation science have called upon public health practitioners and researchers to assess implementation outcomes and incorporate cost analysis into evaluation in order to encourage the translation of research into practice [39]. Researchers in this field have highlighted the importance of identifying and addressing potential barriers to implementation and scale-up and to further understand factors that facilitate adoption at the provider and funder level to improve the acceptability of evidence-based practice and the likelihood of intervention scale-up [6]. Identifying the potential cost-saving benefits of SFL will be an important facilitator toward its scalability. Furthermore, by establishing SFL as a cost-effective health promotion intervention model, this adds further weight to the importance of the partnership approach that underpins SFL and which has been highlighted as a key pillar of its sustainability [12].

Results highlight that this cohort of Shedders rate their dimensions of health relatively positively, resulting in high average utility scores at baseline of 0.827 for the IG and 0.787 for the CG. Research has determined that there is often a discrepancy between men’s objective health measures and how they rate their health subjectively [40]. Moreover, previous studies involving participants both from Sheds and the general population have posited that older people re-calibrate their self-rating of health relative to what they think is reasonable for their age [11,40]. However, when comparing these findings to a comparable study, Men on the Move participants had baseline utility scores of 0.630 in the IG and 0.664 in CG, which are significantly lower than those of Shedders in this study [25]. The difference between Shedders baseline utility scores compared to men in the general community setting may be due to the inherent health-enhancing benefits of the Sheds, which have long been cited in research [7,10,11,16,41]. While the high baseline utility scores arguably make it more difficult for further improvements to be made in terms of benefits derived from SFL, despite this, at 3 months, there was a clear and significant difference for the IG (3.3% improvement) with a further 2% gain at 6 months. This contrasts with an insignificant 1% improvement for the CG over the first 3 months. These improvements in the IG were evident across all of the six dimensions of utility from baseline to 6 months. Although some dimensions did decline from 6 to 12 months, leading to a small decline in utility over this period, utility scores remained significantly higher than baseline at all time-points and notably one year later. Moreover, almost all of the gains achieved from baseline to 6 months were still evident one year later after SFL finished. While there is evidence of sustained improvement overall, this drop-off (which may have been somewhat influenced by COVID-19 restrictions, although not significantly) does highlight the importance of further follow-up with participants in the design and future implementation of SFL to encourage the maintenance of positive behavior change. This is an important consideration and may be indicative of the need for a longer-term evaluation.

From a cost perspective, the total costs of delivering SFL was €130,144 (€309 for each of the 421 participants in the IG), and while it is difficult to compare this on a like-for-like basis to similar studies, this cost per person is shown to be modest and comparable to community-based physical activity interventions for men (Football Fans in training study [42] €239 per participant; Euro FIT [43,44] €221.25 to €312 depending on the country; and Men on the Move [25] €125.82 per participant). Moreover, SFL has a more diverse range of program offerings including but not limited to physical activity, health screenings, healthy eating, mental health, digital literacy, health awareness (cancer, diabetes, dementia, and oral health) and suicide prevention training, which offers an increased level of intervention. When the estimated benefits in the form of improved QALYs are considered, the SFL initiative is shown to generate a cost per QALY that is far below that of established guidelines of €20,000 per QALY [33]. While it should not be assumed that every intervention below the threshold is worth funding if there are cheaper alternatives available, the SFL evaluation is the first economic evaluation of health promotion in Sheds and therefore highlights the benefits of this approach. This gain is reaffirmed through the Sensitivity Analysis, where the probability of success with the intervention is extremely high, even when the costs per QALY exceed its current cost of €15,000.

There are some limitations to this study that should be noted. Firstly, the Sheds operate within a capricious informal environment, which makes a randomized study design unfeasible within this complex real-world system that has many evolving variables. Due to capacity constraints at the time of data collection in Sheds—namely, the availability of two/three data collectors to cover all Sheds and counties as well as the requirement of having to align data collection with Shedder availability and limited Shed opening hours—there were some limitations in terms of the control group and follow-up rates where rescheduling of data collection was not possible. In keeping with the gender-specific approach of SFL, the researchers endeavored to complete all follow-ups in the Shed setting to promote a sense of safety for participants. However, this can present challenges for follow-up rates considering the informality and sporadic attendance in Sheds. Future research may benefit from identifying strategies that would mitigate against this problem, perhaps through hosting an enticing event or the use of other incentives. The control group for this study was a wait list control. Questionnaires were completed in a comparator cohort of Sheds (*n* = 4) due to receive SFL 3 months prior to SFL delivery. This means that a small cohort (*n* = 86) of participants acted as the control and were followed for 3 months only—as these participants transitioned from being the CG to the IG after this period. Moreover, the recruitment of participants into SFL was a sensitive process facilitated by gender-specific approaches where buy-in and trust building is critical to engagement. Therefore, respecting the autonomy of Shedders to opt in/out of the program on their terms took precedence over any attempts to generate a larger size control group. However, research has demonstrated that there is value in having a small control with a larger intervention group in community-based programs where there are often capacity constraints [45]. Indeed, this research calls on researchers to consider an unbalanced design using a relatively small sample size for a control group as it would still improve the amount and quality of available evidence for public health practice and practice-based evidence [45]. The advent of COVID-19 at the time of data collection compounded this difficulty and led to reduced resources, which concentrated on the IG for the remaining time period of the study. The subjective nature of the data and the inherent bias in the self-report format should also be noted, particularly considering the study design where participants are aware they have received an intervention. It is also possible that participants’ self-ratings of health outcomes may have led to some inaccuracies in terms of the benefits that were computed; however, the estimations presented are shown to be still within cost effectiveness thresholds when sensitivity analysis is conducted on the key variables. While the evidence suggests that the recruitment strategy was effective in engaging the target group of Shedders, this approach may lead to a potential selection bias when applied to HTR groups outside of Sheds. Finally, while comparisons can be made between Shedders and the general population of older males in Ireland, SFL is an initiative tailored to the Sheds setting, and therefore, generalizability is limited to the Shedder population.

## 5. Conclusions

This research is the first study that has considered an economic evaluation of men’s health promotion in Sheds. It has highlighted the value in utilizing Sheds as a setting in which to engage men with a targeted health promotion initiative (SFL) that not only has the potential to improve health and well-being outcomes but is also cost effective. The research demonstrates that the partnership design of SFL is an effective way of delivering community-based health initiatives and dispels myths that these approaches are costly. Moreover, findings also further corroborate the value of Sheds as being inherently health enhancing for Shedders. Overall, findings make a valuable contribution to existing research by highlighting the value of community-based men’s health initiatives more broadly in terms of their potential to be cost-effective and health enhancing for men. The results provide a solid evidence base for the future scale-up of SFL and highlight the importance of further research to guide its implementation. Moreover, these findings will be invaluable in advocating for the prioritization of SFL and in the design and delivery of further health promotion initiatives in Shed settings for stakeholders involved in SFL implementation.

## Figures and Tables

**Figure 1 ijerph-19-02204-f001:**
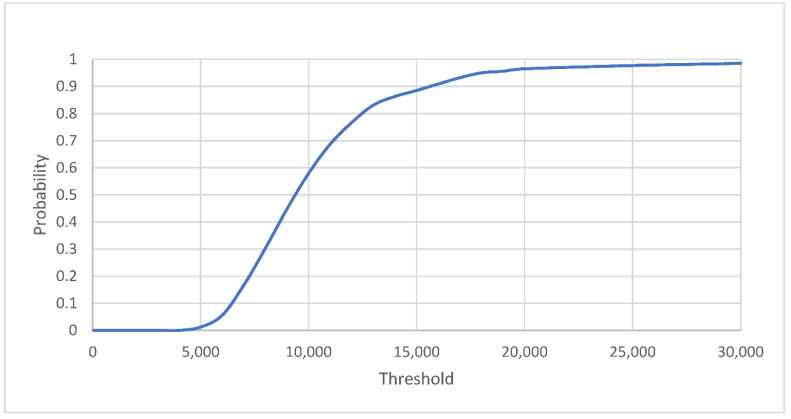
Cost acceptability curve for thresholds up to €30,000.

**Table 1 ijerph-19-02204-t001:** Costs of the Sheds for Life program.

Item	Costs € **			
	Baseline to 3 Months *	3 Months to 6 Months **	6 Months to 12 Months	Overall
Costs of delivering physical activity	13,200	N/A	N/A	
Full health check	25,260	N/A	N/A	
Mental health workshop	9600	N/A	N/A	
Healthy food made easy	19,800	N/A	N/A	
Costs of optional components e.g., digital literacy, cancer awareness, oral health	17,200	N/A	N/A	
Miscellaneous costs e.g., admin costs	212	N/A	N/A	
SFL awards event	0	4546	N/A	
Salary costs including health and wellbeing manager, health administrator	31,863	2278	2278	
Travel and subsistence	2289	809	809	
Total Costs	119,424	7633	3087	130,144
Costs per participant (*n* = 421)				309.13

* Includes costs involved in pre-implementation and implementation phases. ** 3–12 months includes costs involved during maintenance phases. Costs shown in Euro: 1 USD is equivalent to 0.89 Euro.

**Table 2 ijerph-19-02204-t002:** Average value for components of SF-6D over time period by group.

	Baseline			3M			6M		12M	
	IG	CG	Baseline Mean *p*-Value **	IG	CG	3M Mean *p*-Value **	IG	CG	IG	CG
N	379	87		237	75		214	0	266	0
Physical Functioning	2.17	2.28	0.45	1.70	2.23	0.00	1.59	*	1.74	*
Role Limitations	1.49	1.63	0.11	1.32	1.61	0.01	1.24	*	1.24	*
Social Functioning	1.41	1.60	0.06	1.30	1.53	0.02	1.18	*	1.28	*
Pain	2.07	2.13	0.69	1.95	2.12	0.31	1.82	*	1.75	*
Mental Health	2.15	2.22	0.52	1.86	2.27	0.00	1.88	*	1.99	*
Vitality	3.45	3.26	0.10	3.77	3.24	0.00	3.72	*	3.70	*

Key: N = number; M = month; IG = Intervention Group; CG = Comparison-in-waiting Group. * No data for CG ** difference between IG and CG is significant at *p* ≤ 0.05.

**Table 3 ijerph-19-02204-t003:** Utility analysis of the Sheds for Life program.

Group	Baseline	3M	6M	12M
IG-N	374	233	210	260
IG-Average Utility	0.795	0.827 *	0.847 **	0.838 **
Utility Change		0.033	0.020	−0.009
QALYs Gained		7.595	4.222	−2.480
CG-N	85	72	0	0
CG-Average Utility	0.777	0.787		
Utility Change		0.010		
QALYs Gained		0.072		
Program Costs		119,424	7633	
Cost per QALY		15,724	1808	

Key: N = number; M = month; IG = Intervention Group; CG = Comparison-in-waiting Group; QALYs = Quality-Adjusted Life Years; QALYs gained = N × Utility Change. * difference from baseline is significant at *p* ≤ 0.01. ** difference from baseline is significant at *p* ≤ 0.001.

## Data Availability

The data presented in this study are available on request from the corresponding author.

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
