# Peer review of "An Economic Evaluation of ‘Sheds for Life’: A Community-Based Men’s Health Initiative for Men’s Sheds in Ireland"

_ijerph, 2022, doi:10.3390/ijerph19042204_

Round 1
Reviewer 1 Report
The topic of the research proposed by the authors is current and relevant. To authors` knowledge, there has been no other structured health promotion initiatives evaluated in Sheds nor has there been any economic evaluation of health promotion in Sheds. This research is the first study which has considered economic evaluation of men’s health promotion in Sheds. This is the first study to evaluate the cost-effectiveness of health endeavours in Sheds.
The paper is well-structured and easy to follow.
Please find my detailed comments below:
- Introduction
The authors show that the general research area is important and relevant, however, I would recommend some more explicit answers to the following questions:
- why is the paper interesting?
- what is the purpose of it?
- what are the research hypotheses?
- what are the main results?
- Materials and Methods
The research methodology used by the authors is well grounded.
The description of the materials and methods is detailed.
- Results and Discussion
The authors of the research clearly show the results.
The future research directions may also be mentioned.
- Conclusions
Conclusions cover several good points, but they should include the limitation of the research and the future research directions.
- References
Due to the topic of the research, the references are recent.
Regarding the cited publications, the authors must follow the instructions for the references. In the text, reference numbers should be placed in square brackets [ ].
Author Response
Dear reviewer,
The authors would like to sincerely thank you for your constructive comments, conscientious feedback and helpful suggestions for further improving the manuscript. We have considered all suggestions carefully and have prepared a document which responds to all feedback by the reviewers. In this document you will find responses to your review first with subsequent reviewers below. Your review is numbered based on the order it appeared in the submission system.
We hope you find this to your satisfaction and thank you again for taking the time to review the manuscript.
Please see attachment

Reviewer 2 Report
1. What is the purpose of your research? Why does your article indicate no purpose? 2. Your article states that you have substantiated economic efficiency. Please specify how you have calculated it, for what period it is calculated and where can we see these calculations? Also, clarify whether other types of efficiency (social and organizational effectiveness) and aggregate effectiveness were assessed? 3. I have not understood from your article what is its scientific value for theory and practice? What exactly is scientific knowledge incremented in? How can it be evaluated from the standpoint of existing scientific research? 4. Explain the research stages. It would be nice to add a schematic (flow chat) of your research. 5. Who are the users of your research results? Whom will these results be useful for? How can they be used for scientific purposes?
Author Response

(The authors gave the same response as above.)

Reviewer 3 Report
Referee report for “An Economic Evaluation of ‘Sheds for Life’ a Community-Based Men’s Health Initiative for Men’s Sheds in Ireland ”; Manuscript ID: ijerph-1535665, for IJERPH
This is a very interesting research topic. Such initiatives are hugely important since the future of health and healthcare systems is rather grim and community/prevention interventions are a potential solution.
There are some fundamentals of economic evaluation that need to be addressed by the authors. 1) The perspective of the analysis in not mentioned (this affects the whole analysis) 2) The alternative intervention is not clearly stated although I can understand is “do nothing” (other alternatives are not discussed) 3) The time horizon. Since control group stops at 3 months follow-up, I assume it is 3 months but it is not clear. Also, the details of the intervention are limited and another researcher would not be able to reproduce it. Detailed review for each section follows.
Introduction
Since the readership of this journal is international I suggest that the authors add a few words about the Men’s Sheds program early in the paper.
In lines 44-45, it is mentioned that the program has the potential to attract hard-to-reach men as defined by the authors. These individuals may be in a more dire situation health-wise and therefore in a greater need for intervention. I invite the authors to check if these hypotheses are true (perhaps on a future research) or provide literature that supports these claims: 1) hard-to-reach (HTR) men have worse health and 2) men in Sheds have worse health (at baseline) than the respective male population in Ireland or UK (and therefore might be harder-to-reach). I recommend the SHARE-project http://www.share-project.org/home0.html which has a powerful database for the comparative analysis.
In lines 110-111, Ireland has a fundamentally different healthcare system than US (let alone other parameters), I have checked the source but there is no methodology as to how this estimate was obtained. My guess is that this number is borderline arbitrary and should not be used.
Materials and Methods
Regarding the study participants. The wait list control group can provide a similar population sample (of sheders) in theory but it surely limits the generalisability of the study since the no-treatment group is not representative of older males in Ireland. This should be mentioned as a limitation. Moreover, I can see two sources of selection bias, namely the purposive sampling and the expression of interest process. Especially the latter, can filter the sample to a not so HTR men after all, since participants applied to Sheds and volunteered to participate in the research. So the discussion on the HTR needs to be mended.
Regarding data collection. It is not clear if the follow-up starts after the 10 week training. If not, then I doubt that the treatment has the power to objectively alter the health status (and not just perceptions) of the participants in 3 months. Of course this is not a double-blind RT and self-perceived health is precarious (especially with short follow-ups) when subjects know they received treatment. This measurement limitation should be discussed more.
Line 199, were the UK weights used? Can we expect Irish utility weights to be similar?
Line 201, previously undefined acronyms.
Lines 203-205, I feel it is redundant to repeat this information.
Results
Lines 223-225, there is a discrepancy here with lines 184-185 where it states that these costs were NOT included. It is better to separate research costs.
Lines 230-231, I don’t see how this is a cost-effectiveness ratio. Is there an effectiveness denominator? No. It is just a per capita cost.
Table 2. I would like some mean test p-values at least for the baseline groups.
Line 240. The pandemic is a confounder that could skew the results and may explain the diminishing utility as time progresses.
Lines 263-265, the high baseline utility scores are not in favour of the HTR argument based on low-health individuals.
Regarding the sensitivity analysis. What distribution was chosen for the parameters? Were all the parameters simulated? A cost effectiveness plane graph would also be interesting.
Discussion
Lines 319-321, are the previously mentioned researches the general community setting?
Lines 327-328, do the measurements coincide with lockdown measures?
Lines 330-334, with additional re-education costs, a long-term evaluation in necessary.
Lines 343-344, in the context of economic evaluation more intervention is more cost but not necessarily more value, there may be diminishing returns and not a linear dose-response.
Lines 344-348, QALYs discussion should be put in some context. It is not that every intervention below a threshold is worth funding when there are cheaper alternatives. Some examples of the ICERs of other interventions on this population would be great.
Lines 355-356, is there any evidence in the data that the missing follow-ups were not at random?
Author Response

(The authors gave the same response as above.)

Reviewer 4 Report
Congratulations on a nicely written and socially beneficial study. A few minor suggestions.
There could be a couple of stray double spaces in the manuscript. I think the references should be in [] square brackets with no spaces between numbers.
It would be helpful if the introduction could contain a paragraph describing how a shed works, how participants are recruited, how do they get to the shed, how is the site chosen, etc. Give a USD$ equivalent to all costs.
Perhaps give a table listing the QUALY costs for a range of health interventions so that the Sheds QUALY can be seen in a far wider context.
Table 2 can you give the shape of the distributions in an Appendix with a comment on what the shape could imply
Author Response

(The authors gave the same response as above.)

Round 2
Reviewer 2 Report
Thank you for your answers. I am not fully satisfied with the revisions, but I recommend the article for publication nonetheless. Good luck!
Author Response
Thank you for taking the time to review the manuscript so promptly. We greatly appreciate the constructive feedback that improved the work. We endeavored to adequately address all comments and appreciate your recommendation for publication.
Reviewer 3 Report
Dear authors,
I really enjoyed reviewing your work and I wish you the best of luck in future endeavours. Environments like the Sheds and actions like the SFL may be one the best ways to implement preventative geriatric medicine.
I have just two more comments.
Regarding the PSA. If other parameters than the average cost and average QALY were simulated, the normal distribution is inappropriate. Costs are positive and usually right-skewed, hence a Gamma (1.5, 10) will match the parameter better. Utilities also must be bounded between 0 and 1, and so a Beta (2, 3) is preferable. Since I do not know the data used in your analysis, my recommendations are based solely on my experience, so feel to experiment with the distribution parameters.
Regarding the not at random missing follow-ups. I would add a line warning future researchers and encouraging them to find strategies that would mitigate this problem. Off the top of my head, perhaps organise a special event members would not miss, or even a financial incentive.
Author Response
Thank you sincerely for your prompt response. Please see attached.
